# Authenticated Key Exchange Protocol in the Standard Model under Weaker Assumptions

Janaka Alawatugoda [1,2]

1   Research & Innovation Centers Division, Faculty of Resilience, Rabdan Academy,
    Abu Dhabi P.O. Box 114646, United Arab Emirates; jalawatugoda@ra.ac.ae
2   Institute for Integrated and Intelligent Systems, Griffith University, Nathan, QLD 4111, Australia

**Abstract:** A two-party authenticated key exchange (AKE) protocol allows each of the two parties to share a common secret key over insecure channels, even in the presence of active adversaries who can actively control and modify the exchanged messages. To capture the malicious behaviors of the adversaries, there have been many efforts to define security models. Amongst them, the extended Canetti–Krawczyk (eCK) security model is considered one of the strongest security models and has been widely adopted. In this paper, we present a simple construction of a pairing-based eCK-secure AKE protocol in the standard model. Our protocol can be instantiated with a suitable signature scheme (i.e., an existentially unforgeable signature scheme against adaptive chosen message attacks). The underlying assumptions of our construction are the decisional bilinear Diffie–Hellman assumption and the existence of a pseudorandom function. Note that the previous eCK-secure protocol constructions either relied on random oracles for their security or used somewhat strong assumptions, such as the existence of strong-pseudorandom functions, target collision-resistant functions, etc., while our protocol construction uses fewer and more-standard assumptions in the standard model. Furthermore, preserving the same security argument, our protocol can be instantiated with any appropriate signature scheme that comes in the future with better efficiency.

**Keywords:** authenticated key exchange; standard model; eCK model; pairing; weaker assumptions





## 1. Introduction

A two-party key exchange protocol has been a fundamental building block of cryptography and network security. It allows any two parties to share a common session key over an insecure channel. Since its early introduction in 1976, the Diffie–Hellman key exchange protocol [1] has been the most famous key exchange protocol. However, as is well known, the Diffie–Hellman protocol is insecure against the man-in-the-middle attack, where an adversary impersonates one party to the other to read and modify the messages exchanged between two parties. This vulnerability is possible since the parties are not authenticated in the Diffie–Hellman protocol.

To capture such vulnerabilities, there have been many attempts [2–5] to define security models for key exchange protocols in the presence of active adversaries who can actively read and modify the exchanged messages. Among several security models, the extended Canetti–Krawczyk (eCK) model proposed by LaMacchia, Lauter, and Mityagin [5] is considered one of the strongest security models, since it captures various possible behaviors of an active adversary. For instance, the properties captured by the eCK model include the following:

- **Implicit Key Authentication:** If a key exchange protocol provides a guarantee that no party apart from the protocol participants can compute the session key, the key exchange protocol is said to provide *implicit key authentication*. If a key exchange protocol provides implicit key authentication, it is said to be an *authenticated key exchange (AKE)* protocol.

- **Key Confirmation:** If a key exchange protocol provides a guarantee that each party is assured that all other participants possess the same session key, the key exchange protocol is said to provide *key confirmation*.
- **Known Key Security:** The knowledge of a session key should not allow the adversary to learn the session keys in other sessions; all session keys should not depend on the session keys of the other sessions.
- **Security against Unknown Key Share (UKS) Attacks:** Party *A* should not share a session key with party *B*, believing that it is sharing the session key with party *C*. The public keys and identities of the parties should be certified and confirmed or incorporated into protocol execution.
- **Security against Key Compromise Impersonation (KCI) Attacks:** Knowledge of the long-term secret key of party *A* should not enable the adversary to impersonate the other honest parties to the party *A*.
- **(weak) Forward Secrecy:** A (passive) adversary who knows the long-term secret keys of any two parties should not be able to compute the past session keys of the two parties.

Since the proposal of the eCK security model, many eCK-secure AKE protocols have been presented [5–11]. However, some of them [5,6,8,10] are constructed to be secure under the ideal-world assumption of the random oracle model (ROM), and the others are constructed to be secure in the standard model but based on somewhat strong assumptions such as the existence of strong-pseudorandom functions [7] or randomness extractor functions [9].

*Our Contribution*

In this paper, we present a construction of an eCK-secure AKE protocol based on pairings. Our protocol can be instantiated with a suitable signature scheme that is existentially unforgeable against adaptive chosen message attacks. The underlying assumptions of our construction are the decisional bilinear Diffie–Hellman assumption and the existence of a pseudorandom function. Our construction is proven to be secure without the ROM assumption, while many of the existing constructions are proven to be secure in the ROM [12–15]. Apart from that, we remark that our protocol uses fewer and more-standard assumptions compared to the previous standard-model works of Moriyama and Okamoto [7] and Yang et al. [9]. Moreover, preserving the same security argument, our protocol can be instantiated with an appropriate signature scheme that comes in the future with better efficiency; this is also an advantage of our protocol.

## 2. Preliminaries

In this section, we recall the preliminaries that we use in our protocol construction.

### 2.1. Pseudorandom Functions

**Definition 1** (Pseudorandom Function [16]). *Let $F : \{0,1\}^* \times \{0,1\}^* \to \{0,1\}^*$ be an efficient, length-preserving, keyed function. We say F is a pseudorandom function if for all probabilistic polynomial-time adversaries $\mathcal{A}$, there exists a negligible function $\epsilon_{\mathrm{PRF}}$ in the security parameter k such that*

$$\left| \Pr[\mathcal{A}^{F(key,\cdot)}(1^k) = 1] - \Pr[\mathcal{A}^{f_{rnd}(\cdot)}(1^k) = 1] \right| \leq \epsilon_{\mathrm{PRF}},$$

*where $key \in \{0,1\}^k$ is chosen uniformly at random and $f_{rnd}$ is chosen uniformly at random from the set of functions mapping k-bit strings to k-bit strings.*

### 2.2. Existential Unforgeablity Against Adaptive Chosen Message Attacks (EUF-CMA)

**Definition 2** (Existential Unforgeablity Against Adaptive Chosen Message Attacks). *Let $k \in N$ be the security parameter. For a signature scheme $\mathrm{Sig} = (\mathrm{KeyGen}, \mathrm{Sign}, \mathrm{Vfy})$, we define $\mathrm{Adv}^{\mathrm{EUF-CMA}}(\mathcal{B})$ as the advantage of a probabilistic polynomial-time adversary $\mathcal{B}$, winning the following game:*

1. $(sk, vk) \leftarrow \text{KeyGen}(1^k)$
2. $(m^*, \sigma^*) \leftarrow \mathcal{B}^{\mathcal{O}(\cdot)}(vk)$
3. If $\text{Vfy}(vk, m^*, \sigma^*) = $ "*true*" *and* $m^*$ *is not been previously signed, then* $\mathcal{B}$ *wins.*

   <u>*Oracle* $\mathcal{O}(m)$</u>

1. $\sigma \leftarrow (sk, m)$
2. *Return* $\sigma$

   Sig *is* $\text{EUF} - \text{CMA}$ *if* $\text{Adv}^{\text{EUF}-\text{CMA}}(\mathcal{B})$ *is negligible in* $k$.

## 2.3. Decisional Bilinear Diffie–Hellman (DBDH) Assumption

**Definition 3** (Decisional Bilinear Diffie–Hellman Assumption [17])**.** *Let* $k$ *be the security parameter and* $\mathcal{G}$ *be a group generation algorithm. Let* $(\mathbb{G}, \mathbb{G}_T, q, e) \leftarrow \mathcal{G}(1^k)$*, where* $q$ *is a prime number, the description of two groups* $\mathbb{G}, \mathbb{G}_T$ *of order* $q$*, and the description of an admissible bilinear map* $e : \mathbb{G} \times \mathbb{G} \rightarrow \mathbb{G}_T$*. Let* $g, g_1$ *be two arbitrary generators of* $\mathbb{G}$*.*

*The decisional bilinear Diffie–Hellman problem in* $(\mathbb{G}, \mathbb{G}_T, q, e)$ *is as follows: consider two distributions* $(g, g_1, g^a, g^b, e(g, g_1)^{ab})$ *and* $(g, g_1, g^a, g^b, T)$ *for some* $a, b \in \mathbb{Z}_q$ *and random* $T \in \mathbb{G}_T$*. It is said that the DBDH assumption holds in* $(\mathbb{G}, \mathbb{G}_T, q, e)$ *if for all probabilistic polynomial-time algorithms* $\mathcal{A}$*, the advantage of distinguishing the two distributions is given as*

$$\text{Adv}^{\text{DBDH}}_{\mathbb{G}, \mathbb{G}_T, q, e}(\mathcal{A}) = \left| \Pr\left[ \mathcal{A}(g, g_1, g^a, g^b, e(g, g_1)^{ab}) = 1 \right] - \Pr\left[ \mathcal{A}(g, g_1, g^a, g^b, T) = 1 \right] \right|$$

*which is negligible for a given security parameter* $k$*.*

## 3. Extended Canetti-Krawczyk Model

The motivation of LaMacchia et al. [5] in designing the extended Canetti–Krawczyk (eCK) model was that an adversary should have to compromise both the long-term and ephemeral secret keys of a party to recover the session key.

### 3.1. Parties and Long-Term Keys

Let $\mathcal{U} = \{U_1, \ldots, U_{N_P}\}$ be a set of $N_P$ parties. Each party is $U_i$, where $i \in [1, N_P]$ has a pair of long-term public and secret keys, $(pk_{U_i}, sk_{U_i})$. Each party $U_i$ owns at most $N_S$ number of protocol sessions.

### 3.2. Sessions

Each party may run multiple instances of the protocol concurrently or sequentially; we use the term *principal* to refer to a party involved in a protocol instance, and the term *session* to identify a protocol instance at a principal. The notation $\Pi^s_{U,V}$ represents the $s$th session at the owner principal $U$ with the intended partner principal $V$. The principal that sends the first protocol message of a session is the *initiator* of the session, and the principal that responds to the first protocol message is the *responder* of the session. A session $\Pi^s_{U,V}$ enters an *accepted* state when it computes a session key. Note that a session may terminate without ever entering into the accepted state. The information of whether a session has terminated with or without acceptance is public.

### 3.3. Partnering

The legitimate execution of a key exchange protocol between two principals $U$ and $V$ makes two partnering sessions owned by $U$ and $V$, respectively. Two sessions, $\Pi^s_{U,V}$ and $\Pi^{s'}_{U',V'}$, are said to be partners if all of the following hold:

1. Both $\Pi^s_{U,V}$ and $\Pi^{s'}_{U',V'}$ have computed session keys;
2. The messages sent from $\Pi^s_{U,V}$ and the messages received by $\Pi^{s'}_{U',V'}$ are identical;
3. The messages sent from $\Pi^{s'}_{U',V'}$ and the messages received by $\Pi^s_{U,V}$ are identical;
4. $U' = V$ and $V' = U$;

5. Exactly one of $U$ and $V$ is the initiator, and the other is the responder.

The protocol is said to be *correct* if two partner sessions compute identical session keys.

### 3.4. Adversarial Powers

The adversary $\mathcal{A}$ is a probabilistic polynomial-time algorithm in the security parameter $k$ that has control over the whole network. $\mathcal{A}$ interacts with set of sessions that represent protocol instances. $\mathcal{A}$ can adaptively ask the following queries.

- `Send` $(U, V, s, m)$ query: This query allows $\mathcal{A}$ to run the protocol. It sends the message $m$ to the session $\prod_{U,V}^{s}$ as coming from the session $\prod_{V,U}^{s'}$. $\prod_{U,V}^{s}$ will return the next message to $\mathcal{A}$ according to the protocol conversation so far or make a decision on whether to accept or reject the session. $\mathcal{A}$ can also use this query to initiate a new protocol instance with blank $m$. This query captures the capabilities of an active adversary, who can initiate sessions and modify or delay protocol messages.

- `SessionKeyReveal` $(U, V, s)$ query: If a session $\prod_{U,V}^{s}$ has accepted and holds a session key, $\mathcal{A}$ obtains the session key of $\prod_{U,V}^{s}$. A session can only accept a session key once. This query captures the known key attacks.

- `EphemeralKeyReveal` $(U, V, s)$ query: This gives all the ephemeral keys (per session randomness) of the session $\prod_{U,V}^{s}$ to $\mathcal{A}$.

- `Corrupt` $(U)$ query: $\mathcal{A}$ obtains all the long-term secrets of the principal $U$. Then, $\mathcal{A}$ may set up long-term secrets at principal $U$ at will. However, this query does not reveal any session keys to $\mathcal{A}$. This query captures the KCI attacks, UKS attacks and (weak) forward secrecy.

- `Test` $(U, s)$ query: Once a session $\prod_{U,V}^{s}$ has accepted and holds a session key, $\mathcal{A}$ can attempt to distinguish it from a random key. When $\mathcal{A}$ asks the `Test` query, the session $\prod_{U,V}^{s}$ first chooses a random bit $b \in \{0, 1\}$, and if $b = 1$, the actual session key is returned to $\mathcal{A}$; otherwise, a random session key is chosen uniformly at random from the same session key distribution and is returned to $\mathcal{A}$. This query is only allowed to be asked once.

### 3.5. Freshness

A session $\prod_{U,V}^{s}$ is said to be *fresh* if and only if all of the following hold:

1. The session $\prod_{U,V}^{s}$ and its partner (if it exists), $\prod_{V,U}^{s'}$, have not been asked the `SessionKey reveal` query.

2. If the partner $\prod_{V,U}^{s'}$ exists, none of the following combinations have been asked:
   (a) `Corrupt`$(U)$ and `EphemeralKeyReveal`$(U, V, s)$;
   (b) `Corrupt`$(V)$ and `EphemeralKeyReveal`$(V, U, s')$.

3. If partner $\prod_{V,U}^{s'}$ does not exist, none of the following combinations have been asked:
   (a) `Corrupt`$(V)$;
   (b) `Corrupt`$(U)$ and `EphemeralKeyReveal`$(U, V, s)$.

### 3.6. eCK *Security Game*

- Stage 0: The challenger generates the keys by using the security parameter $k$.
- Stage 1: $\mathcal{A}$ is executed and may ask any of the `Send`, `SessionKeyReveal`, `EphemeralKeyReveal`, `Corrupt` queries to any session at will.
- Stage 2: At some point, $\mathcal{A}$ chooses a fresh session and asks the `Test` query.
- Stage 3: $\mathcal{A}$ continues asking `Send`, `SessionKeyReveal`, `EphemeralKeyReveal`,`Corrupt` queries. The only condition is that $\mathcal{A}$ cannot violate the freshness of the test session.
- Stage 4: At some point, $\mathcal{A}$ outputs the bit $b' \in \{0, 1\}$, which is its guess of the value $b$ in the test session. $\mathcal{A}$ wins if $b' = b$.

### 3.7. Definition of Security

Let $\text{Succ}_{\mathcal{A}}$ be the event that the adversary $\mathcal{A}$ wins the eCK game.

**Definition 4** (eCK Security). *A protocol $\pi$ is said to be secure in the* eCK *model if there is no probabilistic polynomial-time adversary $\mathcal{A}$ who can win the* eCK *game with a non-negligible advantage in the security parameter k. The advantage of an adversary $\mathcal{A}$ is defined as:*

$$\mathrm{Adv}_{\pi}^{\mathrm{eCK}}(\mathcal{A}) = |2\Pr(\mathrm{Succ}_{\mathcal{A}}) - 1| \ .$$

### 4. Construction of the Pairing-Based AKE Protocol

We present a pairing-based construction of an eCK-secure AKE protocol, namely protocol EC-P1. Our protocol can be instantiated with any suitable signature scheme (i.e., an existentially unforgeable signature scheme against adaptive chosen message attacks). The security of the protocol EC-P1 is proven in the standard model based on the decisional bilinear Diffie–Hellman assumption and the existence of a pseudorandom function.

*4.1. Protocol Design*

4.1.1. Parameters and Underlying Building blocks

Let $k$ be the security parameter and $\mathcal{G}$ be a group generation algorithm. Let $(\mathbb{G}, \mathbb{G}_T, q, e) \leftarrow \mathcal{G}(1^k)$, where $q$ is a prime number, the description of two groups $\mathbb{G}, \mathbb{G}_T$ of order $q$, and the description of an admissible bilinear map $e : \mathbb{G} \times \mathbb{G} \rightarrow \mathbb{G}_T$. Let $g, g_1$ be arbitrary generators of $\mathbb{G}$ such that $g_1 = g^{\alpha}$ for arbitrary $\alpha \in \mathbb{Z}_q$.

Let $k$ be the security parameter and $\mathrm{Sig} = (\mathrm{KeyGen}, \mathrm{Sign}, \mathrm{Vfy})$ be an $\mathrm{EUF} - \mathrm{CMA}$ signature scheme, where $(\mathrm{KeyGen}, \mathrm{Sign}, \mathrm{Vfy})$ are the key generation, signing and signature verification algorithms, respectively. This protocol uses a signature scheme Sig to sign the protocol messages that are exchanged between the parties.

4.1.2. Initial Setup

Let $a \leftarrow \mathbb{Z}_q$ and $b \leftarrow \mathbb{Z}_q$ be the long-term secret keys of Alice and Bob, respectively, whereas $A \leftarrow g^a$ and $B \leftarrow g^b$ are the long-term public keys of Alice and Bob, respectively. Let $(sk_A, vk_A) \leftarrow \mathrm{KeyGen}(1^k)$ and $(sk_B, vk_B) \leftarrow \mathrm{KeyGen}(1^k)$ be the signing and verification key pairs of the underlying signature scheme $\mathrm{Sig} = (\mathrm{Sign}, \mathrm{Vfy})$ of Alice and Bob, respectively.

4.1.3. Protocol Execution

Let $x$ and $y$ be the ephemeral secret keys of Alice and Bob, respectively, for the current session. Upon picking $x$, Alice computes $W_1 \leftarrow e(A, g_1^x)$. Then, Alice computes the signature $\sigma_A \leftarrow \mathrm{Sign}(sk_A, W_1)$ and sends Alice, Bob, $W_1, \sigma_A$ to Bob over the insecure channel. Upon receipt of Alice, Bob, $W_1, \sigma_A$, Bob picks $y$ and computes $W_2 \leftarrow e(B, g_1^y)$. Then, Bob computes the signature $\sigma_B \leftarrow \mathrm{Sign}(sk_B, W_2)$ and sends Bob, Alice, $W_2, \sigma_B$ to Alice over the insecure channel. After exchanging the public ephemeral values $W_1$ and $W_2$ together with the identities of the corresponding principals and the corresponding signatures over the insecure channel, Alice and Bob verify the signatures $\sigma_A$ and $\sigma_B$, respectively. Then, upon the signature verification, Alice computes $Z_1 \leftarrow e(W_2)^{xa}$, and Bob computes $Z_2 \leftarrow (W_1)^{yb}$. If one of the signatures are not verified, the party will abort the protocol execution. Let PRF be a pseudorandom function. As the final step of the protocol, both Alice and Bob compute the shared key using the pseudorandom function PRF; Alice computes the shared key as $K \leftarrow \mathrm{PRF}(Z_1, \mathrm{Alice}||W_1||\sigma_A||\mathrm{Bob}||W_2||\sigma_B)$, and Bob computes the shared key as $K \leftarrow \mathrm{PRF}(Z_2, \mathrm{Alice}||W_1||\sigma_A||\mathrm{Bob}||W_2||\sigma_B)$.

Note that here, we include all the protocol messages initiated and received by a party at its operation on the pseudorandom function. This is to make sure that no two non-matching sessions compute the same session key. Precisely, as an example, if an adversary reveals a signing key of a protocol principal Alice and re-signs a protocol message from Alice to Bob (such that $\sigma_A \neq \sigma_A'$), even though a verifiable message/signature pair is sent to Bob, the input value used to generate the key $K$ at Alice is different from the input value used to generate the key $K$ at Bob. Therefore, even the adversary is allowed to reveal the

session key at Bob claiming that it is a non-matching session to the session existing at Alice, the session key computed at Alice is different to the session key computed at Bob, and the adversary cannot simply win the security challenge by playing this trick. The execution of the protocol EC-P1 is clearly illustrated in Table 1.

**Table 1.** Protocol EC-P1.

| *Alice* (Initiator) | $\mathbb{G}, \mathbb{G}_T, q, e, g, g_1 \leftarrow \mathcal{G}(1^k)$ | *Bob* (Responder) |
|---|---|---|
| | $\text{Sig} = (\text{KeyGen}, \text{Sign}, \text{Vfy})$ | |
| | **Initial Setup** | |
| $a \leftarrow \mathbb{Z}_q, A \leftarrow g^a$ | | $b \leftarrow \mathbb{Z}_q, B \leftarrow g^b$ |
| $(sk_A, vk_A) \leftarrow \text{KeyGen}(1^k)$ | | $(sk_B, vk_B) \leftarrow \text{KeyGen}(1^k)$ |
| | **Protocol Execution** | |
| $x \leftarrow \mathbb{Z}_q, W_1 \leftarrow e(A, g_1^x)$ | | $y \leftarrow \mathbb{Z}_q, W_2 \leftarrow e(B, g_1^y)$ |
| $\sigma_A \leftarrow \text{Sig}(sk_A, (\text{Alice}, \text{Bob}, W_1))$ | | $\sigma_B \leftarrow \text{Sig}(sk_B, (\text{Bob}, \text{Alice}, W_2))$ |
| | $\xrightarrow{\text{Alice,Bob},W_1,\sigma_A}$ | |
| | $\xleftarrow{\text{Bob,Alice},W_2,\sigma_B}$ | |
| **If** $\text{Vfy}(vk_B, (\text{Bob}, \text{Alice}, W_2), \sigma_B) = \text{"true"}\{$ | | **If** $\text{Vfy}(vk_A, (\text{Alice}, \text{Bob}, W_1), \sigma_A) = \text{"true"}\{$ |
| $Z_1 \leftarrow (W_2)^{xa}$ | | $Z_2 \leftarrow (W_1)^{yb}$ |
| $K \leftarrow \text{PRF}(Z_1, \text{Alice}||W_1||\sigma_A||\text{Bob}||W_2||\sigma_B)$ | | $K \leftarrow \text{PRF}(Z_2, \text{Alice}||W_1||\sigma_A||\text{Bob}||W_2||\sigma_B)$ |
| $\}$ | | $\}$ |
| **else** abort | | **else** abort |
| | $K$ is the session key | |

*4.2. Security Analysis of the Protocol* EC-P1

**Theorem 1.** *Let $k$ be the security parameter and $\mathcal{G}$ be a group generation algorithm. Let $(\mathbb{G}, \mathbb{G}_T, q, e) \leftarrow \mathcal{G}(1^k)$, where $q$ is a prime number, the description of two groups $\mathbb{G}, \mathbb{G}_T$ of order $q$, and the description of an admissible bilinear map $e : \mathbb{G} \times \mathbb{G} \to \mathbb{G}_T$. Let $g, g_1$ be arbitrary generators of $\mathbb{G}$ such that $g_1 = g^\alpha$, where $\alpha \in \mathbb{Z}_q$. If the DBDH assumption holds in $e : \mathbb{G} \times \mathbb{G} \to \mathbb{G}_T$, the function PRF is a pseudorandom function, and the signature scheme Sig is EUF − CMA, then the protocol EC-P1 is secure in the eCK model.*

*Let $\mathcal{U} = \{U_1, \dots, U_{N_P}\}$ be a set of $N_P$ parties. Each party $U_i$ owns at most $N_s$ number of protocol sessions. Let $\mathcal{A}$ be any adversary against the eCK challenger of the protocol EC-P1. Then, the advantage of $\mathcal{A}$ against the eCK security challenge of the protocol EC-P1, $\text{Adv}_{\text{EC-P1}}^{\text{eCK}}$ is:*

$$\text{Adv}_{\text{EC-P1}}^{\text{eCK}}(\mathcal{A}) \leq \max\left(N_P \text{Adv}^{\text{EUF−CMA}}(\mathcal{B}), N_P^2 N_s^2 \left(\text{Adv}_{\mathbb{G}, \mathbb{G}_T, q, e}^{\text{DBDH}}(\mathcal{C}) + \epsilon_{\text{PRF}}\right)\right).$$

*where $\mathcal{C}$ is the algorithm against the DBDH challenger, and $\mathcal{B}$ is the algorithm against the EUF − CMA challenger for the underlying signature scheme Sig. The algorithms $\mathcal{B}$ and $\mathcal{C}$ are constructed using the adversary $\mathcal{A}$ as a subroutine.*

**Proof.** We split the proof of Theorem 1 into two main cases: when the partner to the test session exists and when it does not.

1. A partner to the test session exists.
    (a) The adversary corrupts both the owner and the partner principals to the test session—Case **1a**;
    (b) The adversary corrupts neither the owner nor the partner principal to the test session—Case **1b**;
    (c) The adversary corrupts the owner to the test session but does not corrupt the partner to the test session—Case **1c**;

> (d) The adversary corrupts the partner to the test session but does not corrupt the owner to the test session—Case **1d**;

2. A partner to the test session does not exist: the adversary is not allowed to corrupt the peer to the target session—Case **2**.

We show that the advantage of the adversary $\mathcal{A}$ in each of the above cases is negligible.

*Case **1a**: Adversary Corrupts Both the Owner and Partner Principals to the Test Session*
Game 1

This is the original game. When the Test query is asked, the game 1 challenger will choose a random bit $b \leftarrow \{0,1\}$. If $b = 1$, the real session key is given to $\mathcal{A}$; otherwise, a random value chosen from the same session-key space is given. Hence,

$$\mathrm{Adv}_{\mathrm{Game\,1}}(\mathcal{A}) = \mathrm{Adv}_{\mathrm{EC\text{-}P1,Case\,1a}}^{\mathrm{eCK}}(\mathcal{A}) \ . \tag{1}$$

Game 2

This is the same as game 1 with the following exception: before $\mathcal{A}$ begins, two distinct random principals $U^*, V^* \leftarrow \{U_1, \ldots, U_{N_P}\}$ are chosen, and two random numbers $s^*, t^* \leftarrow \{1, \ldots, N_s\}$ are chosen, where $N_P$ is the number of protocol principals and $N_s$ is the number of sessions on a principal. The session $\Pi_{U^*,V^*}^{s^*}$ is chosen as the target session, and the session $\Pi_{V^*,U^*}^{t^*}$ is chosen as the partner to the target session. If the test session is not the session $\Pi_{U^*,V^*}^{s^*}$ or the partner to the session is not $\Pi_{V^*,U^*}^{t^*}$, the game 2 challenger aborts the game. Unless the incorrect choice happens, game 2 is identical to game 1. Hence,

$$\mathrm{Adv}_{\mathrm{Game\,2}}(\mathcal{A}) = \frac{1}{N_P{}^2 N_s^2} \mathrm{Adv}_{\mathrm{Game\,1}}(\mathcal{A}) \ . \tag{2}$$

Game 3

This is the same as game 2 with the following exception: the game 3 challenger randomly chooses $\delta \leftarrow \mathbb{Z}_q$ and computes $K$ according to the protocol description using $Z_1 = \left(e(g, g_1)^\delta\right)^{ab}$. When the adversary asks the Test$(U^*, V^*, s^*)$ query, the game 3 challenger will answer with $K$.

We construct an algorithm $\mathcal{C}$ against the DBDH challenger using the adversary $\mathcal{A}$ as a subroutine. The game 3 challenger sets all the long-term secret/public key pairs of the protocol principals. The algorithm $\mathcal{C}$ runs a copy of $\mathcal{A}$ and interacts with $\mathcal{A}$ such that $\mathcal{A}$ is interacting with either game 2 or game 3. The DBDH challenger sends values $(g, g_1, g_1^\beta, g_1^\gamma, e(g, g_1)^\delta)$ such that either $\delta = \beta\gamma$ or $\delta \leftarrow \mathbb{Z}_q$ as the inputs to the algorithm $\mathcal{C}$. The game 3 challenger uses $g$ and $g_1$ as the generators for the protocol setup. Moreover, the game 3 challenger computes the value $W_1$ of the target session $(\Pi_{U^*,V^*}^{s^*})$ as $e(A, g_1^\beta)$ and the value $W_2$ of the partner session $(\Pi_{V^*,U^*}^{t^*})$ as $e(B, g_1^\gamma)$. Upon receiving the Test$(U^*, V^*, s^*)$ query, the game 3 challenger computes $K$ using $\left(e(g, g_1)^\delta\right)^{ab}$ as $Z_1$ and answers. The game 3 challenger can answer all the other legitimate queries normally.

If $\mathcal{C}$'s input satisfies $\delta = \beta\gamma$, the simulation constructed by the game 3 challenger is identical to game 2; otherwise, it is identical to game 3. If $\mathcal{A}$ can distinguish the difference between games, then $\mathcal{C}$ can answer the DBDH challenge. Hence,

$$\left|\mathrm{Adv}_{\mathrm{Game\,2}}(\mathcal{A}) - \mathrm{Adv}_{\mathrm{Game\,3}}(\mathcal{A})\right| \leq \mathrm{Adv}_{\mathbb{G},\mathbb{G}_T,q,e}^{\mathrm{DBDH}}(\mathcal{C}) \ . \tag{3}$$

Game 4

This is the same as game 3 with the following exception: the game 4 challenger randomly chooses $K \leftarrow \{0,1\}^k$ and sends it to the adversary $\mathcal{A}$ as the answer to the Test$(U^*, V^*, s^*)$ query.

The game 4 challenger sets all the long-term secret/public key pairs and all the encryption key pairs of the protocol principals. Therefore, the challenger can answer all the queries normally.

If $K$ is computed using the real pseudorandom function with a hidden key, the simulation is identical to game 3, whereas if $K$ is chosen randomly from the session key space, the simulation constructed is identical to game 4. Hence,

$$|\text{Adv}_{\text{Game 3}}(\mathcal{A}) - \text{Adv}_{\text{Game 4}}(\mathcal{A})| \leq \epsilon_{\text{PRF}} \ . \tag{4}$$

Semantic security of the session key in Game 4

Since the session key $K$ of $\Pi^{s^*}_{U^*,V^*}$ is chosen randomly and independently of all other values, $\mathcal{A}$ does not have any advantage in game 4. Hence,

$$\text{Adv}_{\text{Game 4}}(\mathcal{A}) = 0 \ . \tag{5}$$

Using Equations (1)–(5), we find

$$\text{Adv}^{\text{eCK}}_{\text{EC-P1,Case 1a}}(\mathcal{A}) \leq N_P^2 N_s^2 \left( \text{Adv}^{\text{DBDH}}_{\mathbb{G},\mathbb{G}_T,q,e}(\mathcal{C}) + \epsilon_{\text{PRF}} \right) \ .$$

*Case* **1b***: Adversary Corrupts neither the Owner nor the Partner Principals to the Test Session*
Game 1

This is same as game 1 in case 1a. Hence,

$$\text{Adv}_{\text{Game 1}}(\mathcal{A}) = \text{Adv}^{\text{eCK}}_{\text{EC-P1,Case 1b}}(\mathcal{A}) \ . \tag{6}$$

Game 2

This is same as game 2 in case 1a. Hence,

$$\text{Adv}_{\text{Game 2}}(\mathcal{A}) = \frac{1}{N_P^2 N_s^2} \text{Adv}_{\text{Game 1}}(\mathcal{A}) \ . \tag{7}$$

Game 3

Thisi is the same as game 2 with the following exception: the game 3 challenger randomly chooses $\delta \leftarrow \mathbb{Z}_q$ and computes $K$ according to the protocol description using $Z_1 = \left( e(g,g_1)^\delta \right)^{xy}$. When the adversary asks the $\text{Test}(U^*,V^*,s^*)$ query, the game 3 challenger will answer with $K$.

We construct an algorithm $\mathcal{C}$ against the DBDH challenger using the adversary $\mathcal{A}$ as a subroutine. The game 3 challenger sets all the long-term secret/public key pairs of the protocol principals except for the Diffie–Hellman long-term secret keys of the principals $U^*$ and $V^*$ ($a$ and $b$). The algorithm $\mathcal{C}$ runs a copy of $\mathcal{A}$ and interacts with $\mathcal{A}$ such that $\mathcal{A}$ is interacting with either game 2 or game 3. The DBDH challenger sends values $(g, g_1, g^\beta, g^\gamma, e(g,g_1)^\delta)$ such that either $\delta = \beta\gamma$ or $\delta \leftarrow \mathbb{Z}_q$ as the inputs to the algorithm $\mathcal{C}$. The game 3 challenger uses $g$ and $g_1$ as the generators for the protocol setup. For the principal $U^*$, the long-term public key $A$ is set as $g^\beta$, and for the principal $V^*$, the long-term public key $B$ is set as $g^\gamma$. The game 3 challenger computes the $W_1$ of the target session $(\Pi^{s^*}_{U^*,V^*})$ as $e(g^\beta, g_1^x)$ and the $W_2$ of the partner session $(\Pi^{t^*}_{V^*,U^*})$ as $e(g^\gamma, g_1^y)$ by picking $x, y$ at random according to the protocol's specifications. Upon receiving the $\text{Test}(U^*,V^*,s^*)$ query, the game 3 challenger computes the $K$ using $\left( e(g,g_1)^\delta \right)^{xy}$ as $Z_1$ and answers. The game 3 challenger can answer all the other legitimate queries normally.

If $\mathcal{C}$'s input satisfies $\delta = \beta\gamma$, the simulation constructed by the game 3 challenger is identical to game 2; otherwise, it is identical to game 3. If $\mathcal{A}$ can distinguish the difference between games, then $\mathcal{C}$ can answer the DBDH challenge. Hence,

$$|\text{Adv}_{\text{Game 2}}(\mathcal{A}) - \text{Adv}_{\text{Game 3}}(\mathcal{A})| \leq \text{Adv}^{\text{DBDH}}_{\mathbb{G},\mathbb{G}_T,q,e}(\mathcal{C}) \ . \tag{8}$$

Game 4

This is same as game 4 in case 1a. Hence,

$$|\text{Adv}_{\text{Game 3}}(\mathcal{A}) - \text{Adv}_{\text{Game 4}}(\mathcal{A})| \leq \epsilon_{\text{PRF}} . \tag{9}$$

Semantic security of the session key in Game 4

Since the session key $K$ of $\Pi_{U^*,V^*}^{s^*}$ is chosen randomly and independently of all other values, $\mathcal{A}$ does not have any advantage in game 4. Hence,

$$\text{Adv}_{\text{Game 4}}(\mathcal{A}) = 0 . \tag{10}$$

Using Equations (6)–(10), we find

$$\text{Adv}_{\text{EC-P1,Case 1b}}^{\text{eCK}}(\mathcal{A}) \leq N_P^2 N_s^2 \left( \text{Adv}_{\mathbb{G},\mathbb{G}_T,q,e}^{\text{DBDH}}(\mathcal{C}) + \epsilon_{\text{PRF}} \right) .$$

*Case **1c**: Adversary Corrupts the Owner to the Test Session, but Does Not Corrupt the Partner*
Game 1

This is same as hame 1 in case 1a. Hence,

$$\text{Adv}_{\text{Game 1}}(\mathcal{A}) = \text{Adv}_{\text{EC-P1,Case 1c}}^{\text{eCK}}(\mathcal{A}) . \tag{11}$$

Game 2

This is same as game 2 in case 1a. Hence,

$$\text{Adv}_{\text{Game 2}}(\mathcal{A}) = \frac{1}{N_P^2 N_s^2} \text{Adv}_{\text{Game 1}}(\mathcal{A}) . \tag{12}$$

Game 3

This is the same as game 2 with the following exception: the game 3 challenger randomly chooses $\delta \leftarrow \mathbb{Z}_q$ and computes $K$ according to the protocol description using $Z_1 = \left( e(g,g_1)^\delta \right)^{ya}$. When the adversary asks the $\text{Test}(U^*, V^*, s^*)$ query, the game 3 challenger will answer with $K$.

We construct an algorithm $\mathcal{C}$ against the DBDH challenger using the adversary $\mathcal{A}$ as a subroutine. The game 3 challenger sets all the long-term secret/public key pairs of the protocol principals except for the Diffie–Hellman long-term secret key of the principal $V^*$ ($b$). The algorithm $\mathcal{C}$ runs a copy of $\mathcal{A}$ and interacts with $\mathcal{A}$ such that $\mathcal{A}$ is interacting with either game 2 or game 3. The DBDH challenger sends values $(g, g_1, g^\beta, g^\gamma, e(g,g_1)^\delta)$ such that either $\delta = \beta\gamma$ or $\delta \leftarrow \mathbb{Z}_q$ as the inputs to the algorithm $\mathcal{C}$. The game 3 challenger uses $g$ and $g_1$ as the generators for the protocol setup. For the principal $V^*$, the long-term public key $B$ is set as $g^\beta$. Moreover, the game 3 challenger computes the value $W_1$ of the target session ($\Pi_{U^*,V^*}^{s^*}$) as $e(A, g^\gamma)$. Upon receiving the $\text{Test}(U^*, V^*, s^*)$ query, the game 3 challenger computes the $K$ using $\left( e(g,g_1)^\delta \right)^{ya}$ and answers, where $y$ is chosen at random according to the protocol specification. The game 3 challenger can answer all the other queries normally.

If $\mathcal{C}$'s input satisfies $\delta = \beta\gamma$, the simulation constructed by the game 3 challenger is identical to game 2; otherwise, it is identical to game 3. If $\mathcal{A}$ can distinguish the difference between games, then $\mathcal{C}$ can answer the DBDH challenge. Hence,

$$|\text{Adv}_{\text{Game 2}}(\mathcal{A}) - \text{Adv}_{\text{Game 3}}(\mathcal{A})| \leq \text{Adv}_{\mathbb{G},\mathbb{G}_T,q,e}^{\text{DBDH}}(\mathcal{C}) . \tag{13}$$

Game 4

This is same as game 4 in case 1a. Hence,

$$|\mathrm{Adv}_{\mathrm{Game\ 3}}(\mathcal{A}) - \mathrm{Adv}_{\mathrm{Game\ 4}}(\mathcal{A})| \le \epsilon_{\mathrm{PRF}} \ . \tag{14}$$

Semantic security of the session key in Game 4

Since the session key $K$ of $\Pi_{U^*,V^*}^{s^*}$ is chosen randomly and independently of all other values, $\mathcal{A}$ does not have any advantage in game 4. Hence,

$$\mathrm{Adv}_{\mathrm{Game\ 4}}(\mathcal{A}) = 0 \ . \tag{15}$$

Using Equations (11)–(15), we find

$$\mathrm{Adv}_{\mathrm{EC\text{-}P1,Case\ 1c}}^{\mathrm{eCK}}(\mathcal{A}) \le N_P^2 N_s^2 \left( \mathrm{Adv}_{\mathbb{G},\mathbb{G}_T,q,e}^{\mathrm{DBDH}}(\mathcal{C}) + \epsilon_{\mathrm{PRF}} \right) \ .$$

*Case* **1d***: Adversary Corrupts the Partner to the Test Session, but Does Not Corrupt the Owner*

The analysis of this case is similar the analysis of case 1c. The only difference is in game 3. We briefly explain the simulation of game 3 as follows:

We construct an algorithm $\mathcal{C}$ against the DBDH challenger using the adversary $\mathcal{A}$ as a subroutine. The game 3 challenger sets all the long-term secret/public key pairs of the protocol principals except for the Diffie–Hellman long-term secret key of the principals $U^*$ ($a$). The algorithm $\mathcal{C}$ runs a copy of $\mathcal{A}$ and interacts with $\mathcal{A}$ such that $\mathcal{A}$ is interacting with either game 2 or game 3. The DBDH challenger sends values $(g, g_1, g^\beta, g^\gamma, e(g, g_1)^\delta)$ such that either $\delta = \beta\gamma$ or $\delta \leftarrow \mathbb{Z}_q$ as the inputs to the algorithm $\mathcal{C}$. The game 3 challenger uses $g$ and $g_1$ as the generators for the protocol setup. For the principal $U^*$, the long-term public key $A$ is set as $g^\beta$. Moreover, the game 3 challenger computes the value $W_2$ of the partner session ($\Pi_{V^*,U^*}^{t^*}$) as $g^\gamma$. Upon receiving the $\mathtt{Test}(U^*, V^*, s^*)$ query, the game 3 challenger computes $K$ using $\left(e(g, g_1)^\delta\right)^{xb}$ and answers, where $x$ is chosen at random according to the protocol specification. The game 3 challenger can answer all the other queries normally.

Apart from the foregoing changes in the game 3 simulation, the rest of the simulation of case 1d is the same as case 1c. Therefore, we obtain

$$\mathrm{Adv}_{\mathrm{EC\text{-}P1,Case\ 1d}}^{\mathrm{eCK}}(\mathcal{A}) \le N_P^2 N_s^2 \left( \mathrm{Adv}_{\mathbb{G},\mathbb{G}_T,q,e}^{\mathrm{DBDH}}(\mathcal{C}) + \epsilon_{\mathrm{PRF}} \right) \ .$$

*Case* **2***: Partner to the Test Session Does Not Exist*

In this case, there is no partner that exists for the target session. Note that the owner of the target session is $U^*$. In this case, the peer session is supposed to be at the principal $V^*$, but the peer session does not exist at $V^*$. When there is no peer session existing, the adversary itself is involved in computing the protocol message as the partner of the target session. Note that the adversary is not allowed to corrupt the peer principal $V^*$, and the adversary does not have the signing key of $V^*$.

Game 1

This is same as game 1 in case 1a. Hence,

$$\mathrm{Adv}_{\mathrm{Game\ 1}}(\mathcal{A}) = \mathrm{Adv}_{\mathrm{EC\text{-}P1,Case\ 2}}^{\mathrm{eCK}}(\mathcal{A}) \ . \tag{16}$$

Game 2

Before A begins, the game 2 challenger guesses the identity, $V^*$, of the partner principal to the test session, and if the guess is incorrect, it aborts the game. The probability of game 2 being aborted due to an incorrect guess of the partner principal to the test session is $1 - \frac{1}{N_P}$. Unless the incorrect guess happens, game 2 is identical to game 1. Hence,

$$\text{Adv}_{\text{Game 2}}(\mathcal{A}) = \frac{1}{N_P} \text{Adv}_{\text{Game 1}}(\mathcal{A}) \ . \tag{17}$$

The algorithm $\mathcal{B}$ sets the verification key of the $\text{EUF} - \text{CMA}$ signature scheme challenger to the principal $V^*$. The owner principal accepts the message coming from the intended partner if the owner computes $\text{Vfy}(vk_{V^*}, (V^*, U^*, W_2), \sigma_{V^*}) = $ "true". However, the principal $V^*$ is not corrupted, and the message $(V^*, U^*, W_2)$ is not signed by the principal $V^*$, because there is no partner to the test session. Hence,

$$\text{Adv}_{\text{Game 2}}(\mathcal{A}) = \text{Adv}_{\text{Sig}}^{\text{EUF}-\text{CMA}}(\mathcal{B}) \ . \tag{18}$$

Using Equations (16)–(18), we find

$$\text{Adv}_{\text{EC-P1,Case 2}}^{\text{eCK}}(\mathcal{A}) = N_P \text{Adv}_{\text{Sig}}^{\text{EUF}-\text{CMA}}(\mathcal{B}) \ .$$

*Combining All the Above Cases*

According to the analysis, we can see that the adversary $\mathcal{A}$'s advantage of winning against the eCK challenger of the protocol EC-P1 is:

$$\text{Adv}_{\text{EC-P1}}^{\text{eCK}}(\mathcal{A}) \leq \max \big( N_P \text{Adv}^{\text{EUF}-\text{CMA}}(\mathcal{B}),$$
$$N_P^2 N_s{}^2 \big( \text{Adv}_{\mathbb{G},\mathbb{G}_T,q,e}^{\text{DBDH}}(\mathcal{C}) + \epsilon_{\text{PRF}} \big) \big) \ .$$

where $\mathcal{C}$ is the algorithm against the DBDH challenger and $\mathcal{B}$ is the algorithm against the $\text{EUF} - \text{CMA}$ challenger for the underlying signature scheme Sig. The algorithms $\mathcal{B}$ and $\mathcal{C}$ are constructed using the adversary $\mathcal{A}$ as a subroutine. $\quad \square$

### 4.3. Computational Costs

We provide the overall computational cost of our protocol at a protocol principal (either the initiator or the responder) in Table 2. Note that the costs for light computations such as multiplication/division are ignored. In general, the AKE protocols in the standard model require more computational costs compared to those in the ROM.

**Table 2.** Overall computational cost at a protocol principal.

| | Operation<br>*At the Initiator or the Responder* | Computational Cost |
|---|---|---|
| **Initial setup** | Computation of $A$ or $B$<br>Signature key generation | 1E<br>KeyGen |
| **Protocol execution** | Computation of the protocol message<br>Computation of $Z_1$ or $Z_2$<br>Computation of $K$ | 1Pair, 1E, Sign<br>Vfy, 1E, 1Pair<br>1PRF |

Key: PRF—pseudorandom function; E—exponentiation; KeyGen—key generation function of the signature scheme; Sign—signing function; Vfy—signature verification function of the signature scheme; Pair—pairing.

We provide a comparison of our protocol with several existing eCK-secure AKE protocols in Table 3. Generally, one pairing calculation is approximately four times slower than one modulo exponentiation [18]. Moreover, multiple modulo exponentiation is almost as efficient as single modulo exponentiation [19]. The overhead of computing PRFs is minimal.

**Table 3.** Basic characteristics of few eCK-secure AKE protocols.

| Protocol | Proof Model | Hardness Assumptions | Overall Computational Cost *At a Protocol Principal* |
|---|---|---|---|
| NAXOS [5] | ROM | GDH | 4E |
| CMQV [8] | ROM | GDH | 3E |
| KFU P1 [6] | ROM | GDH | 3E |
| KFU P2 [6] | ROM | CDH | 5E |
| ASB [10] | ROM | GDH | 6E |
| TFNS19 [12] | ROM | XDH, $q$-gap | 5H, 1Pair, 6E |
| Daniel et al. [13] | ROM | GDH | 5PM |
| Xie et al. [14] | ROM | GDH | 4PM |
| Lian et al. Type-II [15] | ROM | Gap-BDH | 4CR, 1Pair, 4E, 1KeyGen, 1Enc, 1Dec, 1KDF |
| Lian et al. Type-III [15] | ROM | Gap-BDH | 5CR, 1Pair, 5E, 1KeyGen, 1Enc, 1Dec, 1KDF |
| MO [7] | Standard | DDH, CR, $\pi$PRF | 3E, 2CR, 1ME, 1$\pi$PRF |
| Yang P1 [9] | Standard | DBDH, PRF, TCR | 2E, 4ME, 4Pair, 2TCR, 1PRF |
| Yang GC-KKN [9] | Standard | DDH, TCR, PRF, FAC, EXT | 7E, 2ME, 2TCR, 3PRF |
| Bergsma et al. Protocol II [20] | Standard | PRF | 16E, 12Pair, 4PRF, 1KeyGen, 1Sign, 1Vfy, 1NIKEgen, 4NIKEkey |
| EC-P1 (this paper) | Standard | DBDH, PRF | 3E, 2Pair, 1PRF, 1KeyGen, 1Sign, 1Vfy |

Key: ROM—random oracle model; GDH—gap DH; CDH—computational DH; DDH—decisional DH; DBDH—decisional bilinear DH; XDH—external DH; $q$-gap—$q$-gap-bilinear collision attack assumption; PRF—pseudorandom function; $\pi$PRF—strong-pseudorandom function; CR—collision resistant function; TCR—target collision resistant function; EXT—randomness extractor function; H—hash function; KDF—key derivation function; KeyGen—key generation function of the signature scheme/authenticated encryption scheme; Sign, Vfy—respectively signing and signature verification functions of the signature scheme; Enc, Dec—respectively encryption and decryption functions of the authenticated encryption scheme; NIKEgen—key generation function of the non-interactive key exchange protocol; NIKEkey— key evaluation function of the non-interactive key exchange protocol; FAC—factorization; E—exponentiation; ME—multi-exponentiation; Pair—pairing; PM—elliptic curve point multiplication.

In the paper of Dutta et al. [21], there is a comparison between existing pairing-based cryptographic protocols, which compares existing pairing-based signature schemes as well. According to that, we see the short signature scheme of Boneh and Boyen [22] is a good candidate to instantiate our protocol. This signature scheme is proven to be secure in the standard model. It requires $2n$ scalar multiplications in $\mathbb{G}_1$ for KeyGen, 1 inversion in $\mathbb{Z}_q$ and 1 scalar multiplications in $\mathbb{G}_1$ for Sign, 2 multiplications in $\mathbb{G}_1$, 2 additions in $\mathbb{G}_1$, and 2 pairing computations, one of which one can be pre-computed for Vfy.

## 5. Conclusions and Future Works

Usually, the AKE protocols that are proven to be secure in the standard model require strong assumptions to achieve the eCK security. We construct a standard model eCK-secure AKE protocol based on pairings. We emphasize that we use fewer and more-standard assumptions compared to the previous works. Thus, our contribution is a valuable improvement in the context of key exchange protocols. As a future work, we can implement this protocol to be used with the TLS protocol suite. Moreover, it is worthwhile to research quantum-safe and leakage-resilient AKE protocols.

**Funding:** APC is supported by the Rabdan Academy, UAE.

**Data Availability Statement:** Not applicable.

**Conflicts of Interest:** The author declares no conflict of interest.

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
