# Peer review of "Authenticated Key Exchange Protocol in the Standard Model under Weaker Assumptions"

_cryptography, doi:10.3390/cryptography7010001_

Round 1

Author Response

Thank you for your detailed report.

Reviewer 2 Report

In this paper,  the author designed a new authenticated key exchange protocol, and  proved the security of this new protocol in the standard model. The assumption underlying this construction is the decisional bilinear Diffie-Hellman assumption and the existence of a pseudorandom function.  
This submission is pretty interesting. Because the required time for this review is short, I did not get enough time to check the proof details.
I guess this submission is OK for publication in this journal.

Page 1, abstract, line -7: Note that the previous ...
Page 5, line -1: Alice computes the shared
Page 12, reference [2] and [3]: no conference here

Author Response

Thank you

Round 2

Author Response

Thank you very much for your thorough comments to improve the quality of our paper.
